# Gradually Compacting Large Language Models for Reasoning Like a Boiling Frog

## Abstract

Large Language Models (LLMs) have demonstrated impressive reasoning capabilities, but their substantial size often demands significant computational resources. To reduce resource consumption and accelerate inference, it is essential to eliminate redundant parameters without compromising performance. However, conventional pruning methods that directly remove such parameters often lead to a dramatic drop in model performance in reasoning tasks, and require extensive post-training to recover the lost capabilities. In this work, we propose a gradual compacting method that divides the compression process into multiple fine-grained iterations, applying a Prune–Tune Loop (`PTL`) at each stage to incrementally reduce model size while restoring performance with finetuning. This iterative approach—reminiscent of the "boiling frog" effect—enables the model to be progressively compressed without abrupt performance loss. Experimental results show that `PTL` can compress LLMs to nearly half their original size with only lightweight post-training, while maintaining performance comparable to the original model on reasoning tasks. Moreover, `PTL` is flexible and can be applied to various pruning strategies, such as neuron pruning and layer pruning, as well as different post-training methods, including continual pre-training and reinforcement learning. Additionally, experimental results confirm the effectiveness of `PTL` on a variety of tasks beyond mathematical reasoning, such as code generation, demonstrating its broad applicability.

## 1 Introduction

Large language models (LLMs) (OpenAI, 2023; Grattafiori et al., 2024; Team et al., 2023; 2024; Yang et al., 2024) have demonstrated remarkable reasoning capabilities, achieving state-of-the-art performance across a wide range of NLP tasks, including multi-step planning (Wang et al., 2024; Hsiao et al., 2025; Wei et al., 2025), tool use (Shi et al., 2025; Qu et al., 2025; Luo et al., 2025), collaboration in multi-agent settings (Tran et al., 2025; Guo et al., 2024), code generation (Jiang et al., 2024; Huang et al., 2025), and facilitating scientific discovery (Zhang et al., 2024b; Chen et al., 2025). However, the remarkable reasoning performance of LLMs often comes with a trade-off, as they typically consist of billions of parameters and require substantial computational resources (Goldstein et al., 2023; Musser, 2023). To mitigate these challenges, researchers have proposed various approaches to reduce model size, such as knowledge distillation (Xu et al., 2024; Gu et al., 2024; Zhang et al., 2025b), pruning (Ma et al., 2023; Men et al., 2024), and matrix approximation (Sy et al., 2024; Ashkboos et al., 2024a). Nonetheless, most of these model compression methods result in unstructured models (Ma et al., 2023; Ashkboos et al., 2024a) or require extensive post-training to recover performance after compression (Ma et al., 2023; Ashkboos et al., 2024a; Men et al., 2024).

In this work, we propose a novel Prune-Tune Loop (`PTL`) method, designed to gradually compact large language models through lightweight post-training, while preserving their original reasoning capabilities. As shown in Figure 1, we divide the entire compacting process into multiple iterations, each consisting of two steps: *pruning*, to remove parameters redundant to reasoning, and *tuning*, to restore the model's reasoning performance. By ensuring that each step introduces only minor changes, any performance degradation is minimal and can be quickly recovered through lightweight training, akin to the "boiling frog" effect where gradual changes avoid abrupt disruptions. Specifically, during the pruning step, we focus on removing either neurons or layers that are redundant for reasoning. A neuron refers to a column or row in the parameter matrices, while a layer corresponds to an entire Transformer block (Vaswani et al., 2017), including both the attention mechanism and the feed-

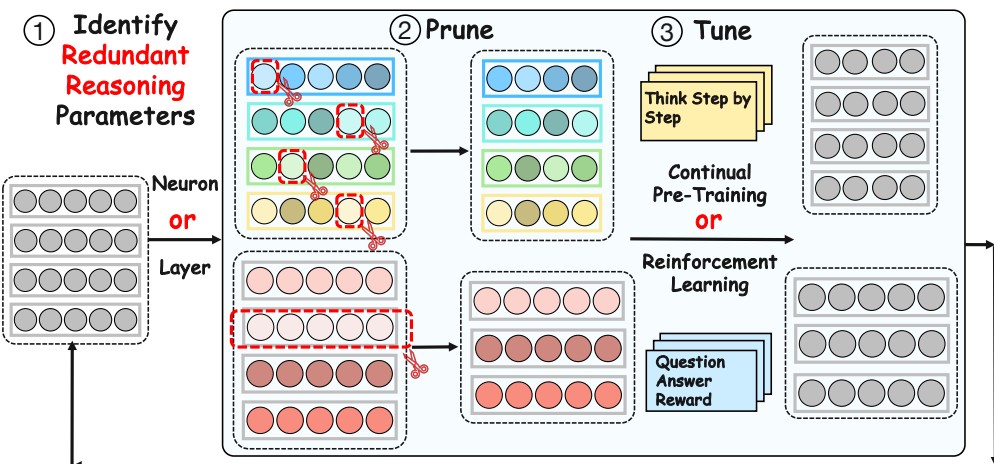

Figure 1: Prune–Tune Loop (PTL) mainly consists of three steps in each minimal iteration: ① identify redundant reasoning parameters, including either redundant neurons or layers; ② prune the identified redundant reasoning parameters; ③ tune the pruned model through either continual pre-training on CoT data or reinforcement learning using complex mathematical datasets. By introducing only minor changes at each iteration, performance degradation remains minimal and quickly recoverable, avoiding abrupt disruptions like the "boiling frog" effect.

forward network. Furthermore, a neuron or layer is considered redundant if its removal has minimal effect on the embeddings produced by the LLM when processing reasoning-related sequences. Meanwhile, during the tuning step, we apply either continual pre-training using Chain-of-Thought (CoT) data (Wei et al., 2022) or reinforcement learning with complex mathematical data (Shao et al., 2024) to recover model's reasoning capability.

We conduct comprehensive experiments to evaluate the performance of PTL across various models and tasks. Our compression method was evaluated on three widely used open-source models, Llama3-8B (Grattafiori et al., 2024), Qwen2.5-7B (Yang et al., 2024), and Gemma2-9B (Team et al., 2024), and we demonstrate that it can reduce each model to approximately 60% of its original size without any loss in mathematical reasoning capability. Specifically, Llama3-8B is compressed from 8 billion to 5 billion parameters, yielding a 30% reduction in FLOPs and a 224% increase in overall runtime efficiency, while maintaining comparable accuracy on GSM8K (Cobbe et al., 2021) (from 54.7% to 52.5%), Minerva Math (Hendrycks et al., 2021) (from 16.0% to 18.5%), and MATH-500 (Lightman et al., 2023) (from 14.6% to 16.1%). Likewise, Gemma2-9B is pruned from 9 billion to 5 billion parameters, halving its FLOPs (50% of the original) and boosting runtime efficiency to 130%, with performance essentially unchanged on GSM8K (from 70.0% to 70.6%), Minerva Math (from 29.1% to 26.4%), and MATH-500 (from 26.9% to 26.2%). Furthermore, on the Llama3 compression benchmark our approach outperforms all alternative baselines, and on the Gemma2 benchmark it is the only method that preserves near-original performance after aggressive pruning. For the Qwen2.5-7B model, we explored reinforcement learning-based approaches to recover performance after compression. Experimental results show that only the model obtained via PTL is able to restore performance comparable to that of the original model on GSM8K (from 85.7% to 84.9%), Minerva Math (from 26.1% to 24.1%) and MATH-500 (from 61.0% to 61.2%). In contrast, other compressed variants consistently fail to recover using reinforcement learning, i.e., performance is 0 on all datasets. In addition, experimental results confirm the effectiveness of PTL on a variety of tasks beyond mathematical reasoning. On code generation tasks, we evaluate our method on Llama3-8B, compressed the model to 5 billion parameters, and the performance on the MBPP benchmark (Austin et al., 2021) with a slight drop (from 50% to 45%), while having 30% reduction in FLOPs and speed up the model 2.56 times.

## 2 PRUNE-TUNE LOOP

To compact a large language model for reasoning, we aim to identify and remove redundant parameters that do not contribute to its reasoning ability, and subsequently post-tune the model to restore any

lost performance. In this section, we introduce a prune-tune loop as an efficient compacting method that minimizes additional training cost.

## 2.1 REDUNDANT REASONING PARAMETER EXTRACTION

Without compromising generalizability, and inspired by prior pruning studies (Xia et al., 2023; Men et al., 2024; Muralidharan et al., 2024), we primarily adopt two structured pruning approaches: removing redundant neurons and pruning redundant layers.

**Redundant Reasoning Neuron Extraction** We define each neuron as one column or one row of a parameter matrix and identify neurons that are NOT activated when LLMs doing reasoning tasks. Formally, we denote each reasoning-related input sequence as $x$, and suppose the model has $L$ layers. Then the embedding and layerwise forward pass can be expressed as

$$h_0(x) = \text{Embed}(x), \quad h_\ell(x) = f_\ell\big(h_{\ell-1}(x)\big), \quad \ell = 1, \ldots, L, \tag{1}$$

where $f_\ell(\cdot)$ denoted the parameter of LLM's $\ell$-th layer. A neuron $\mathcal{N}$ in layer $\ell$ is considered not activated if its removal significantly alters the output of the layer. Specifically, we regard $\mathcal{N}$ as not activated if

$$\big\|f_\ell\big(h_{\ell-1}(x)\big) - f_{\ell\setminus\mathcal{N}}\big(h_{\ell-1}(x)\big)\big\|_2 \leq \sigma, \tag{2}$$

where $\sigma$ is a predefined threshold, and $f_{\ell\setminus\mathcal{N}}(\cdot)$ denotes the layer's output with neuron $\mathcal{N}$ removed.

Prior studies (Hou et al., 2023; Zhao et al., 2024) indicate that reasoning capabilities in large language models are primarily attributed to the self-attention mechanism, suggesting that redundant reasoning neurons rarely appear in this component. Additionally, the self-attention layer contains significantly fewer parameters than the feed-forward layer (Grattafiori et al., 2024; Team et al., 2024), limiting its potential for compression. Furthermore, pruning neurons in the self-attention layer is structurally constrained, as it requires maintaining a consistent number of neurons across all attention heads or eliminating entire heads (Ma et al., 2023). Therefore, we focus on pruning the feed-forward structure rather than the self-attention mechanism.

Specifically, in the circumstance of neurons in the feed-forward structure, Equation 2 can be equivalently transferred as

$$\begin{aligned}
&\big\|f_\ell\big(h_{\ell-1}(x)\big) - f_{\ell\setminus\mathcal{N}}\big(h_{\ell-1}(x)\big)\big\|_2 \\
&= \big\|\text{FFN}_\ell(x) - \text{FFN}_{\ell\setminus\mathcal{N}}(x)\big\|_2 \propto \text{SiLU}\big(W_{\text{gate}}(x)\big) \cdot W_{\text{up}}(x) := \text{Act}_\ell^{\mathcal{N}}(x),
\end{aligned} \tag{3}$$

where $\text{FFN}_\ell(\cdot)$ denotes the feed-forward network of the $\ell$-th layer, while $W_{\text{up}}$ and $W_{\text{gate}}$ correspond to the up-projection and gating matrices, respectively. In other words, the change in the embedding is proportional to the activation of the feed-forward network. Therefore, when inputting reasoning-related input sequence $x$, redundant reasoning neurons are extracted through

$$\mathcal{R}_{\mathcal{N}} := \{\mathcal{N} \mid \text{Act}_\ell^{\mathcal{N}}(x) \leq \sigma_{\text{neuron}}, \ \forall x, \ \ell = 1, \ldots, L\}. \tag{4}$$

**Redundant Reasoning Layer Extraction** In addition to pruning redundant reasoning neurons, we adopt a structured layer pruning approach that identifies layers with minimal contribution to the reasoning task. Formally, following Equation 2, the importance of a layer is quantified by the change it induces in the embedding (Zhang et al., 2024a; Men et al., 2024). Specifically, the importance of layer $\ell$ is measured by the $L_2$ norm of the difference between its output and input embeddings, i.e.,

$$\big\|f_\ell\big(h_{\ell-1}(x)\big) - h_{\ell-1}(x)\big\|_2. \tag{5}$$

Therefore, when inputting reasoning-related input sequence $x$, redundant reasoning layers are extracted through

$$\mathcal{R}_{\mathcal{L}} := \{\ell \mid \big\|f_\ell\big(h_{\ell-1}(x)\big) - h_{\ell-1}(x)\big\|_2 \leq \sigma_{\text{layer}}, \ \forall x, \ \ell = 1, \ldots, L\}. \tag{6}$$

---

**Algorithm 1** Prune–Tune Loop

---

**Input:** Original reasoning language model $\mathcal{LLM}$, reasoning task sequences $x$, thresholds $\sigma_{\mathrm{neuron}}, \sigma_{\mathrm{layer}}$, CoT training data $\mathcal{D}_{CoT}$, RL training data $\mathcal{D}_{RL}$ max pruning rounds $T$.

1: // Compact the model until reach maximal rounds.
2: **while** $t < T$ **do**
3:   // Extract redundant parameters.
4:   Extract redundant reasoning neurons $\mathcal{R}_{\mathcal{N}}$ through Equation 4.
5:   Extract redundant reasoning layers $\mathcal{R}_{\mathcal{L}}$ through Equation 6.
6:   // Remove redundant parameters.
7:   $\mathcal{LLM} \leftarrow \mathcal{LLM} \ominus \{\mathcal{R}_{\mathcal{N}} \cup \mathcal{R}_{\mathcal{L}}\}$
8:   // Recovery Tuning.
9:   $\mathcal{LLM} \leftarrow \text{Continual Training}(\mathcal{LLM}, \mathcal{D}_{CoT})$
10:   $\mathcal{LLM} \leftarrow \text{Reinforcement Learning}(\mathcal{LLM}, \mathcal{D}_{RL})$
11: **end while**
**Output:** Compressed model $\mathcal{LLM}$

---

## 2.2 RECOVERY TUNING

After removing redundant reasoning parameters, the model's architecture is altered, which can result in performance degradation across various tasks. To help the model's parameters adapt to the modified structure, post-training is essential to recover its original capabilities. We primarily employ continual pre-training—a widely used recovery approach following model editing (Ma et al., 2023; Muralidharan et al., 2024; Zhao et al., 2025). In addition, we leverage reinforcement learning, commonly adopted in reasoning models, to further enhance the model's reasoning abilities (Guo et al., 2025; Zhang et al., 2025a).

**Continual Pre-Training**  To restore the performance of the pruned model on reasoning tasks, we perform post-training using a reasoning-focused corpus. Specifically, we utilize math questions accompanied by corresponding CoT reasoning path (Wei et al., 2022), covering a wide range of mathematical scenarios.

**Reinforcement Learning**  Similar to RL training approach used with the base model (Zeng et al., 2025), we apply GRPO (Shao et al., 2024) to further train the model on more challenging math problems after pruning. Additionally, we adopt both standard format reward and accuracy reward, without relying on complex reward design or elaborate control mechanisms.

## 2.3 PRUNE-TUNE LOOP

To enhance the effectiveness of pruning and enable a higher pruning ratio without requiring extensive post-training, we propose a prune-tune loop for gradually compacting LLMs. The pruning process is divided into multiple iterations, with model performance recovered at each step. Specifically, each iteration consists of the following cycle: (1) extraction of redundant reasoning parameters, (2) removal of redundant parameters, and (3) recovery fine-tuning. This prune-tune loop follows a gradual, iterative approach—reminiscent of the "boiling frog" effect—that enables the model to be progressively compressed, ultimately yielding a model with the desired parameter scale. Algorithm 1 provides s detailed illustration of the prune-tune loop compacting algorithm.

## 3 EXPERIMENT

In this section, we evaluate the proposed method on its efficacy and the important factors through extensive studies.

Table 1: Main results of `PTL` on LLaMA3-8B, Gemma2-9B, and Qwen2.5-7B. The number of FLOPs and speedup are computed as averages over all questions in the respective datasets. All methods use identical training data and hardware environments for recovery. Recovery time is measured in hours. Baseline methods that fail to recover in the RL setting are denoted with "-" in the table.

| Method | GSM8K | | | Minerva Math | | | MATH-500 | | | Recovery |
|---|---|---|---|---|---|---|---|---|---|---|
| | Accu. | #FLOPs | Speedup | Accu. | #FLOPs | Speedup | Accu. | #FLOPs | Speedup | |
| *Continual Pre-Training Recovery* | | | | | | | | | | |
| **Llama3-8B** | 54.7 | 9.1 T | 1.0 | 16.0 | 8.5 T | 1.0 | 14.6 | 8.5 T | 1.0 | 0 |
| Shortgpt | 42.7 | 6.4 T | 2.1 × | 15.1 | 6.0 T | 1.8 × | 15.6 | 6.0 T | 1.8 × | 16 H |
| Slicegpt | 45.5 | **4.5** T | 2.1 × | 14.7 | **4.1** T | 1.6 × | 14.1 | **4.1** T | 1.6 × | 20 H |
| Prune-Once | 44.6 | 5.8 T | **2.6** × | 14.2 | 5.4 T | **2.1** × | 14.7 | 5.4 T | **2.1** × | **12** H |
| `PTL-Llama3-5B` | 52.5 | 6.3 T | **2.6** × | **18.5** | 5.9 T | **2.1** × | **16.1** | 5.9 T | **2.1** × | **12** H |
| **Gemma2-9B** | 70.0 | 11.5 T | 1.0 | 29.1 | 10.5 T | 1.0 | 26.9 | 10.4 T | 1.0 | 0 |
| Shortgpt | 55.3 | 7.6 T | **1.6** × | 19.8 | 6.8 T | **1.4** × | 19.3 | 6.7 T | **1.4** × | 27 H |
| Slicegpt | 56.3 | **5.9** T | 1.5 × | 18.6 | **5.3** T | 1.3 × | 19.1 | **5.2** T | 1.4 × | 36 H |
| Prune-Once | 58.7 | 6.5 T | 1.3 × | 18.9 | 6.0 T | 1.3 × | 18.2 | 5.9 T | 1.2 × | **20** H |
| `PTL-Gemma2-5B` | 70.6 | 6.1 T | 1.3 × | **26.4** | 5.6 T | 1.3 × | **26.2** | 5.5 T | 1.2 × | **20** H |
| *Reinforcement Learning Recovery* | | | | | | | | | | |
| **Qwen2.5-7B** | 85.7 | 9.0 T | 1.0 | 26.1 | 8.2 T | 1.0 | 61.0 | 8.2 T | 1.0 | 0 |
| Shortgpt | 0 | - | - | 0 | - | - | 0 | - | - | - |
| Slicegpt | 0 | - | - | 0 | - | - | 0 | - | - | - |
| Prune-Once | 0 | - | - | 0 | - | - | 0 | - | - | - |
| `PTL-Qwen2.5-5B` | **84.9** | 4.9 T | 1.2 × | **24.1** | 4.5 T | 1.4 × | 61.2 | 4.5 T | 1.4 × | **64** H |

## 3.1 EXPERIMENT SETUP

**Datasets** We primarily utilize two large-scale mathematical reasoning datasets: NuminaMath-CoT (LI et al., 2024), which contains 860k diverse math problems ranging from high school exercises to international olympiad-level questions formatted in a CoT style, and MetaMathQA (Yu et al., 2023), comprising 390k problem-solution pairs enhanced through various data augmentation techniques to promote diverse and robust reasoning pathways. These datasets are merged into a unified corpus for redundant reasoning parameter extraction, where each question is concatenated with its corresponding CoT answer to form a complete reasoning sequence. The same dataset is employed during the tuning stage for recovery through continual pre-training. Additionally, when reinforcement learning is used for recovery, we adopt Math-12k[1], a challenging math dataset derived from (Lightman et al., 2023).

**Backbone Models** We evaluate three widely-used open-source LLMs as backbone models. Llama3-8B (Grattafiori et al., 2024), Qwen2.5-7B (Yang et al., 2024) and Gemma2-9B (Team et al., 2024).

**Baselines.** We employ several state-of-the-art pruning methods as baselines and apply the same post-training data and training procedure to the models after pruning. (i) ShortGPT (Men et al., 2024) directly deletes the redundant layers in LLMs based on an importance score; (ii) SliceGPT (Ashkboos et al., 2024b) replaces each weight matrix with a smaller dense matrix, reducing the embedding dimension of the network; (iii) Prune-Once removes redundant parameters in a single step and recovers the model using the same method as `PTL`.

**Benchmarks** To assess mathematical reasoning proficiency, we employ three benchmarks: GSM8K (Cobbe et al., 2021), which contains 8.5k grade-school word problems requiring 2–8 steps of basic arithmetic; Minerva Math (Hendrycks et al., 2021), which fine-tunes PaLM (Chowd-hery et al., 2022) on mathematical and scientific texts to produce LaTeX-formatted solutions in

---

[1] `https://huggingface.co/datasets/hiyouga/math12k`

a few-shot setting; and MATH-500 (Lightman et al., 2023), a 500-question subset of the MATH benchmark curated for rigorous, decontaminated evaluation.

**Evaluation Metrics** We use accuracy to evaluate the performance of the models on each dataset. To assess the impact of model compaction, we also consider the number of floating-point operations (FLOPs), which represents the total count of floating-point arithmetic operations, as well as the speedup achieved. Additionally, we measure the post-training time to quantify the computational cost associated with recovering the model's reasoning capabilities after compression.

**Implement Details** For the continual pre-training recovery setting, we employed the LLaMA-Factory library (Zheng et al., 2024), a widely adopted GitHub-hosted framework for efficient large-model fine-tuning, to carry out all training procedures. Experiments are conduced on four 80GB NVIDIA A100 GPUs, with learning rate as $8 \times 10^{-6}$, global batch size as 32, and max token as 1024. Furthermore, to reduce memory consumption during training, we applied ZeRO Stage-2 optimization and gradient checkpointing, both provided by the DeepSpeed library. For reinforcement learning training, we use the EasyR1 (Zheng et al., 2025) framework built on verl (Sheng et al., 2024), with specialized support for VLMs. Experiments are conducted using eight 140GB NVIDIA H200 GPUs with a global batch size of 128, a rollout batch size of 128, a rollout temperature of 1.0, a consistent learning rate of $1 \times 10^{-6}$, and 8 rollouts. Additionally, to mitigate the risk of model overfitting caused by repeated exposure to the same corpus, we uniformly partitioned the dataset into multiple non-overlapping subsets of equal size.

## 3.2 MAIN RESULTS

Table 1 presents the performance of the compressed models obtained via `PTL`, alongside the original models and those produced by baselines.

**`PTL` consistently outperforms other compacting methods.** As shown by accuracy, although our approach achieves a substantial reduction in parameter count (around $40\%$), it maintains capabilities largely comparable to the unpruned models and outperforms other compression methods by a clear margin. For GSM8K, PTL-Llama3-5B falls from $54.7\%$ to $52.5\%$ after pruning the last two layers and 7000 neurons per layer. By contrast, Shortgpt-Llama3-5B and Slicegpt-Llama3-5B score just $42.7\%$ and $45.5\%$, respectively. On Minerva Math, PTL-Llama3-5B achieves $18.5\%$ vs. $16.0\%$ originally. By comparison, other 5B base-

Table 2: Results of applying RL training to compacted Qwen2.5-7B model in the Continual pre-Training (CT) setting.

| Model | GSM8K | Minerva |
|---|---|---|
| **Qwen2.5-7B** | 85.7 | 26.1 |
| CT-Qwen2.5-5B | 70.9 | 15.8 |
| $\hookrightarrow$ + RL-Zero | **86.9** | **34.2** |

lines linger around $14$–$15\%$. On MATH-500, the accuracy of PTL-Llama3-5B rose from $16.0\%$ in the original model to $18.9\%$, representing a greater improvement than that achieved by the two alternative pruning methods. For the Gemma2-9B, only our PTL-Gemma2-5B maintains performance close to that of the original model across all three benchmarks (GSM8K: from $70.0\%$ to $70.6\%$; Minerva Math: from $29.1\%$ to $26.4\%$; MATH-500: from $26.9\%$ to $26.2\%$), whereas alternative pruning techniques incur severe accuracy losses (GSM8K: from $70.0\%$ to $51.4\%/56.3\%$; Minerva Math: from $29.1\%$ to to $19.8\%/18.6\%$; MATH-500: from $26.9\%$ to $19.3\%/19.1\%$).

For Qwen2.5-7B, we explore the use of reinforcement learning to recover the model's performance after compression. Notably, our PTL-Qwen2.5-7B is the only pruned model that is able to recover performance comparable to—or even surpassing—that of the original model on GSM8K (from $85.7\%$ to $84.9\%$), Minerva Math (from $26.1\%$ to $24.1\%$), and MATH-500 (from $61.0\%$ to $61.2\%$). In contrast, models compressed using alternative pruning methods consistently fail during RL fine-tuning, often producing incoherent or invalid outputs. This clearly demonstrates the scalability and robustness of our pruning methodology. Moreover, we observe that models in the Qwen family perform exceptionally well on several widely used mathematical reasoning benchmarks, making it difficult to recover their original performance using only open-source datasets. However, as shown in Table 2 by applying reinforcement learning on top of models processed through `PTL` under continual pre-training setting, we are able to restore—and in some cases even surpass—the performance of the original model (GSM8K: from $85.7\%$ to $86.9\%$, Minerva Math: from $26.1\%$ to $34.2\%$).

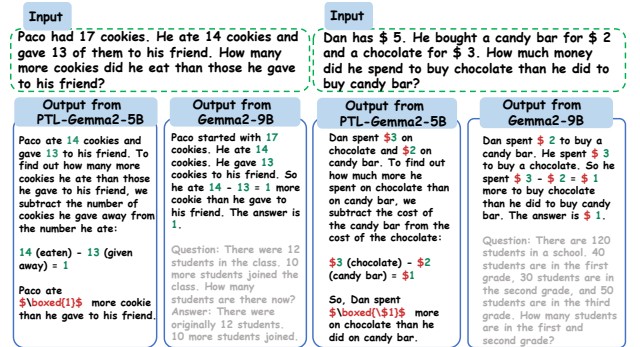 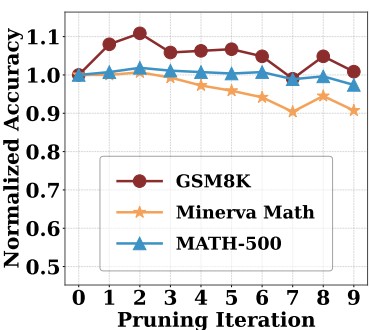

Figure 2: PTL-Gemma2-5B generates more structured CoT and avoids producing meaningless output.

Figure 3: PTL-Gemma2-9B's performance over a complete pruning cycle

**PTL can speedup and requires minimal recovery time.** In contrast to simplistic "pseudo-compression" approaches that merely skip certain layers or modify inter-layer data flow patterns, our pruning methodology achieves genuine model acceleration through substantive parameter reduction that streamlines the data generation pipeline. This structural optimization enables significant computational acceleration in the model's processing workflow. In terms of FLOPs, as shown in Table 1, our PTL-Llama3-5B reduces computational cost by 30% relative to the original model, yielding an overall runtime efficiency of 224% across the three datasets—surpassing Shortgpt-Llama3-5B (188%) and Slicegpt-Llama3-5B (175%). Moreover, PTL-Gemma2-5B is the only model that maintains its performance after aggressive pruning, with FLOPs reduced to 53% of the original and runtime efficiency increased to 127%.

**PTL is a lightweight and user-friendly solution.** Unlike some pruning methods that produce uneven parameter distributions and complicate deployment, our approach maintains consistent per-layer parameter counts and adheres to the Hugging Face format, ensuring ease of use. It requires minimal changes to the model's `generate` function and keeps pruning logic separate from the model architecture. This modularity improves scalability, broadens applicability across diverse models, and removes framework-specific constraints, enabling seamless deployment without sacrificing functionality.

In addition, we evaluate the pruned model on SQuAD (Rajpurkar et al., 2016), a commonsense reasoning benchmark. The results are shown in Appendix A.1. Interestingly, although the models were only recovered using the math reasoning task, their performance on other reasoning tasks—including SQuAD—remains well-preserved, demonstrating the generalization ability of the compacted models.

### 3.3 CONCRETE EXAMPLES

To provide a general understanding of our compacted model, Figure 2 presents several concrete examples comparing its outputs with those of the original model for the same questions. We find that through the incorporation of CoT datasets in our training pipeline, the pruned model has acquired enhanced reasoning capabilities while maintaining output structures consistent with our training data format. For example, `PTL-Gemma2-5B` generates a more structured CoT, as demonstrated by more formal equations and well-structured answers enclosed in `\box{}`. It also avoids producing meaningless output, which is labeled in gray in the figure..

## 4 FURTHER ANALYSIS

### 4.1 ABLATION ANALYSIS

In this section, we present a more detailed analysis of the pruning process, examining it from three principal perspectives: pruning iterations, pruning step size, and pruning order.

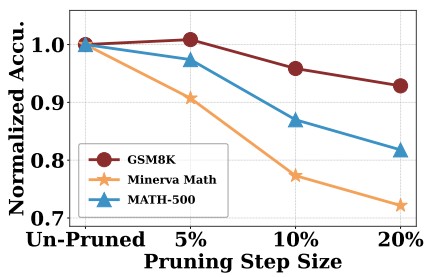

Figure 4: The impact of varying pruning step size on the model's final performance.

Table 3: Impact of the relative ordering of pruning methods over a full pruning cycle on the final model performance. "Lay-Neu" indicates pruning layers first, followed by neurons. "Re-Neu-Lay" denotes alternating between neuron and layer pruning. "Neu-Lay" refers to pruning neurons first, then layers.

| | Orig. | Lay-Neu | Re-Neu-Lay | Neu-Lay |
|---|---|---|---|---|
| **GSM8K** | 70.0 | 69.1 | 68.7 | 70.6 |
| **Minerva Math** | 29.1 | 24.1 | 24.3 | 26.4 |
| **MATH-500** | 26.9 | 25.4 | 23.0 | 26.2 |

**Pruning Iterations**  To rigorously evaluate the effectiveness and stability of our pruning methodology, we meticulously documented the capability evolution of the Gemma2-9B model throughout an entire pruning cycle, as illustrated in Figure 3. The results reveal a notable phenomenon: during the initial pruning iterations, the model's performance exhibited measurable improvement (GSM8K: from 70.0% to 77.6%; Minerva Math: from 29.1% to 29.3%; MATH-500: from 26.9% to 27.4%) rather than degradation, attributable to the complementary fine-tuning process. Subsequently, as parameter removal progressed, the model's capability gradually declined before eventually stabilizing at a level comparable to the original model.

**Pruning Step Size**  To validate the stability of our pruning methodology, we conducted three experimental trials with varying pruning step sizes (5%, 10%, and 20%) while maintaining identical target model sizes. The experimental results, presented in Figure 4, demonstrate an inverse correlation between pruning step size and final model performance - smaller pruning steps (5%) consistently yield better preserved accuracy than larger steps (10%, 20%). Based on the empirical results, the model pruned with 5% step achieves an average accuracy across the three datasets that is 9% higher than the model pruned at 10% step and 14% higher than the model pruned at 20% step. However, this performance advantage comes at the cost of requiring more iterative pruning cycles to achieve the target model size, presenting a practical trade-off between computational efficiency and model quality that practitioners need to consider.

**Pruning Order**  To systematically evaluate the stability of our pruning methodology, we conducted three experimental trials applying distinct pruning sequences (neuron-then-layer, layer-then-neuron, and alternating neuron-layer pruning) while maintaining identical compression ratios. As demonstrated in Table 3, the neuron-then-layer pruning sequence achieved best model performance. While the other two approaches showed marginally inferior results, the performance differences were statistically small. For these three models, the maximum difference in average accuracy across the three datasets is only 6%. This consistent performance across all pruning sequences robustly validates the architectural stability of our method.

In addition, we compare training the full model with training the pruned model at each iteration of `PTL` to investigate whether the training data enhances reasoning capability rather than merely aiding recovery. Further details are provided in Appendix A.2.

## 4.2 MORE TASK

**Experiment Settings**  Besides mathematical reasoning tasks, we also explore `PTL` on coding datasets. For this task, we use opc-sft-stage2 (subset of OpenCoder (Huang et al., 2025)) and the python subset of StarCoder (Li et al., 2023) for neuron-level code generation ability probing and as pretraining corpora. For language ability probing, we used the same text corpus as in the previous experiment to maintain consistency and ensure comparability. To evaluate the compacted model's ability on coding tasks, we adopt MBPP (Austin et al., 2021), offering a set of about $1k$ problems specifically designed for code generation. The experiment follows the typical pruning setting from the previous experiment, and uses Llama3-8B as the backbone models here.

**Main Result** Table 4 summarizes the result of `PTL` under Llama3-8B (Grattafiori et al., 2024). We can observe that our method also shows good performance after rounds of Prune-Tune Loop. Specifically, the accuracy on MBPP (with three shots) only dropped about 5% after pruning the last two layers and 5000 neurons on

Table 4: Result of `PTL` on MBPP (3-shot) under Llama3-8B.

| Method | Accu. | #FLOPs | Speedup | Recovery |
|---|---|---|---|---|
| **Llama3-8B** | 50.0 | 2.8 T | 1.0 | 0 |
| Shortgpt-Llama3-5B | 11.7 | 2.0 T | 2.1 × | 14 H |
| `PTL-Llama3-5B` | **45.0** | **1.9 T** | **2.6 ×** | **12 H** |

each remaining layer (over 30% pruning ratio). Overall, `PTL` excels in performance and accuracy on coding tasks after certain pruning ratio.

## 5 RELATED WORK

**LLM Reasoning** LLMs have demonstrated outstanding performance across a broad spectrum of NLP applications, including multi-step reasoning (Wang et al., 2024; Hsiao et al., 2025), tool use (Shi et al., 2025; Qu et al., 2025), and collaboration in multi-agent settings (Tran et al., 2025; Guo et al., 2024). Multi-step reasoning has gained considerable attention, with models such as QimProving (Wang et al., 2024) and recent work by Hsiao et al. (2025) exploring how LLMs can perform complex tasks requiring logical inference over multiple steps. These systems demonstrate an impressive ability to plan, adapt, and resolve ambiguity across different domains, from natural language understanding to complex decision-making tasks. Tool use is another area in which LLMs have shown great potential. Recent work (Shi et al., 2025) has demonstrated how LLMs can learn to interact with external tools, such as APIs and databases, to enhance their capabilities. Furthermore, collaboration in multi-agent environments has been explored as a key aspect of LLMs' reasoning capabilities. Recent surveys (Tran et al., 2025) and studies (Guo et al., 2024) highlight the ability of LLMs to function as part of a collaborative multi-agent system, where agents must coordinate and communicate to solve problems. In the realm of code generation, LLMs have revolutionized the way software is developed. Models like OpenCoder (Huang et al., 2025) have shown SOTA performance in automatically generating and understanding code. Lastly, LLMs have played an important role in scientific discovery.

**LLM Compression** LLMs, with billions of parameters, demand substantial computational resources for training and inference (Goldstein et al., 2023; Musser, 2023), making deployment in constrained environments challenging. To address this, various compression strategies have emerged. Knowledge distillation trains a smaller "student" model to replicate a larger "teacher" model, achieving size reductions with minimal performance loss (Xu et al., 2024; Gu et al., 2024; Fang et al., 2025; Lee et al., 2025; Zhang et al., 2025b). Pruning removes less important parameters or neurons, effectively reducing model size (Ma et al., 2023; Men et al., 2024; Xia et al., 2023). Matrix approximation techniques like low-rank factorization further compress models by approximating weight matrices, offering promising gains in size and inference speed (Sy et al., 2024; Ashkboos et al., 2024a). However, these methods often yield unstructured models or require significant post-training fine-tuning to recover performance (Ma et al., 2023; Ashkboos et al., 2024a; Men et al., 2024). Balancing compression with performance remains a key challenge.

## 6 CONCLUSION

In this work, we propose a progressive model compression framework that iteratively prunes redundant parameters during the training loop. Our method achieves a compression ratio of 30% to 40%, resulting in inference speedups ranging from 30% to 160%. Compared to baseline models, our approach offers not only superior performance but also greater ease of use. Furthermore, we evaluate our method on a code-related benchmark and show that it can be effectively transferred to the domain of code understanding and generation. We provide a detailed exposition of the design principles underlying our method and conduct extensive experiments to comprehensively analyze its effectiveness. Our results confirm the method's stability and effectiveness across a wide range of hyperparameter settings.

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

## LLM USAGE

We used LLMs as general-purpose writing and debugging assistants. Specifically, LLMs were employed to help polish the writing (e.g., improving sentence clarity, grammar, and flow) and occasionally to assist with debugging minor implementation issues (e.g., identifying syntax errors or suggesting code refactoring). However, all core ideas, research questions, methodological designs, codebase implementations, experiments, and analyses were entirely conceived, developed, and conducted by the authors. No part of the intellectual contribution, experimental framework, or scientific reasoning was generated by an LLM.

## LIMITATION

`PTL` has two primary limitations. First, we do not employ the performance on instruct-following tuned model, as it does not guarantee that the model's ability to follow instructions is fully preserved after compression. Second, the current implementation only supports open-source models. Nevertheless, its simplicity and efficiency make it promising for fast inference and lightweight deployment.

## A   APPENDIX

### A.1   MORE TASKS

Table 5: Exact Match and F1 scores for different models and their PTL-pruned versions.

|  | Gemma2-9B | PTL-Gemma2-5B | Llama3-8B | PTL-Llama3-5B |
|---|---|---|---|---|
| **Exact Match** | 27.4 | 28.2 | 27.3 | 37.2 |
| **F1** | 34.8 | 32.9 | 33.6 | 40.6 |

We evaluate the pruned model on SQuAD, a commonsense reasoning benchmark. The results are shown below. Interestingly, although the models were only recovered using the math reasoning task, their performance on other reasoning tasks—including SQuAD—remains well-preserved, demonstrating the generalization ability of the compacted models.

### A.2   TRAINING FULL MODEL

Table 6: Comparison of reasoning performance and model size between full model training (CoT Full) and PTL pruned model training across iterations based on Gemma2-9B.

| CoT Full Model | | | | PTL Pruned Model | | | |
|---|---|---|---|---|---|---|---|
| **Method** | **GSM8K** | **MATH** | **Size** | **Method** | **GSM8K** | **MATH** | **Size** |
| Original | 70.0 | 29.6 | 9.2B | Original | 70.0 | 29.6 | 9.2B |
| CoT Full 1 | 77.6 | 30.0 | 9.2B | PTL Iteration 1 | 75.6 | 29.0 | 8.8B |
| CoT Full 2 | 78.3 | 31.3 | 9.2B | PTL Iteration 2 | 77.6 | 29.4 | 8.4B |
| CoT Full 3 | 77.2 | 31.0 | 9.2B | PTL Iteration 3 | 74.1 | 28.9 | 7.9B |
| CoT Full 4 | 78.1 | 31.3 | 9.2B | PTL Iteration 4 | 74.4 | 28.3 | 7.5B |
| CoT Full 5 | 77.1 | 31.6 | 9.2B | PTL Iteration 5 | 74.7 | 27.9 | 7.0B |
| CoT Full 6 | 75.7 | 32.0 | 9.2B | PTL Iteration 6 | 73.4 | 27.4 | 6.6B |
| CoT Full 7 | 78.0 | 32.5 | 9.2B | PTL Iteration 7 | 69.3 | 26.3 | 6.2B |
| CoT Full 8 | 75.1 | 31.1 | 9.2B | PTL Iteration 8 | 73.3 | 27.5 | 5.8B |

We conduct a comparison between training the full model and training the pruned model in each iteration of `PTL` based on Gemma2-9B. Table 6 shows that training on the full dataset indeed improves performance, demonstrating the validity and high quality of the dataset. However, we also observe

that the pruned model in each `PTL` iteration achieves performance that is comparable to the full model, despite having a significantly smaller size. This highlights the effectiveness of our pruning method, which maintains competitive performance while reducing model complexity.

