# OpenReview forum: "Gradually Compacting Large Language Models for Reasoning Like a Boiling Frog"
_ICLR.cc/2026/Conference — Submitted to ICLR 2026_

### Official Review · Reviewer_N6qf · 2025-10-28

**Soundness:** 2
**Presentation:** 2
**Contribution:** 2
**Rating:** 6
**Confidence:** 3

**Summary:**

This paper addresses the challenge of compressing Large Language Models while preserving their reasoning capabilities, where standard pruning methods often fail, leading to significant performance degradation. The authors propose a iterative compression method called the Prune-Tune Loop (PTL). The core idea is to avoid abrupt performance collapse by applying gradual changes. The authors conduct extensive experiments on modern LLMs (Llama3-8B, Gemma2-9B, Qwen2.5-7B) and show that PTL can reduce model size by 40-50% while maintaining performance on math and code domain

**Strengths:**

1. The proposed PTL method provides effective solution that achieves substantial FLOPs reduction (30-50%) and measured speedup (up to 2.6x) with minimal performance loss.

2. The paper is well-written. The PTL method is introduced with a clear diagram, simple mathematical formulations, and a clean algorithm.

**Weaknesses:**

1. The metrics used to identify redundant parameters are standard and relatively simple. While their effectiveness is proven by the results, the paper could be strengthened by a detailed discussion justifying this choice and the reason why it can help for perserve reasoning capability.

2. The pruning model are finetuned using reasoning-focused corpus, which is straightforward to recover performance and implies the pruning is guided by performance on this specific domain. How sensitive is the method to this dataset across different domains? Does this "reasoning-focused" cause disproportionate harm to other, non-reasoning capabilities? A brief discussion on the trade-offs of using a specialized dataset for pruning versus a general-purpose corpus would be valuable.

**Questions:**

The result that all baseline pruned models fail (0% accuracy) during RL recovery is striking. Do you have a hypothesis for this? Is it that the large, one-shot structural changes from baseline pruning move the model's policy into a region of the parameter space from which RL (GRPO) cannot recover, whereas PTL's gradual changes keep the policy within a stable rajectory?

---

> ### Author Response · Authors · 2025-11-24
> **Response to Reviewer N6qf**
>
> Dear Reviewer N6qf,
>
> Thank you for your insightful reviews and comments. We appreciate the time and effort you have put into providing valuable feedback. We would like to address your concerns as follows:
>
> > Concern #1: General corpus
>
> We evaluate the pruned model on SQuAD, a commonsense reasoning benchmark. The results are shown below. Interestingly, although the models were only recovered using the math reasoning task, their performance on other reasoning tasks—including SQuAD—remains well-preserved, demonstrating the generalization ability of the compacted models.
>
> |             | Gemma2-8B | PTL-Gemma2-5 | Llama3-8B | PTL-Llama3-5B |
> | ----------- | --------- | ------------ | --------- | ------------- |
> | Exact Match | 27.4      | 28.2         | 27.3      | 37.2          |
> | F1          | 34.8      | 32.9         | 33.6      | 40.6          |
>
> Regarding the use of other detection corpora, we agree that employing a more general corpus can help preserve broader general capabilities. However, this often comes at the cost of reasoning performance, as the reasoning-specific capabilities are not explicitly targeted during pruning. This highlights an inherent trade-off between generality and reasoning specialization, which we plan to investigate further in future work.
>
> ---
>
> > Concern #2: 0% for baselines in RL setting.
>
> This happens because the model completely fails, meaning it cannot generate any meaningful responses. As a result, the RL training process also fails.

---

### Official Review · Reviewer_kkhn · 2025-10-28

**Soundness:** 2
**Presentation:** 3
**Contribution:** 1
**Rating:** 4
**Confidence:** 3

**Summary:**

This paper proposes Prune-Tune Loop (PTL), a gradual model compression method for Large Language Models that maintains thier capabilities while reducing model size by 30-40%. The key idea of the paper is dividing the compression process into multiple iterations, where each iteration removes a small portion of redundant parameters followed by lightweight recovery training using either continual pre-training or reinforcement learning. Experiments on Llama3-8B, Qwen2.5-7B, and Gemma2-9B demonstrate that PTL can compress models to ~60% of their original size while maintaining comparable performance on mathematical reasoning benchmarks (GSM8K, Minerva Math, MATH-500) and code generation tasks.

**Strengths:**

1. The paper demonstrates impressive compression ratios (30-40% parameter reduction) with minimal performance loss across multiple state-of-the-art models. Particularly notable is the Gemma2-9B result where PTL is the only method maintaining near-original performance after aggressive pruning.
2. The evaluation spans three different model architectures, multiple mathematical reasoning benchmarks, and includes code generation tasks. The ablation studies examining pruning iterations, step sizes, and ordering provide valuable insights.
3. The formalization of redundant neuron and layer identification is mathematically precise, with clear equations defining the importance metrics based on activation patterns and embedding changes.

**Weaknesses:**

1. The core contribution is essentially iterative application of existing techniques (magnitude-based pruning, importance scoring via activation norms, and recovery fine-tuning). The "Prune-Tune Loop" is not fundamentally different from gradual magnitude pruning approaches that have been explored in the literature.
2. The paper lacks any theoretical analysis. There are no convergence guarantees, compression bounds, or formal analysis of why multiple small pruning steps outperform single large steps. Prior work in optimization and neural network theory could provide relevant frameworks that the authors ignore.
3. This work is primarily an engineering exercise that combines well-known techniques without advancing our theoretical understanding of LLM pruning and recovery. The paper does not provide new insights into how LLMs encode reasoning capabilities or why gradual removal preserves them better.
4. No comparison with iterative magnitude pruning, gradual sparsity methods, or recent structured pruning techniques that use similar multi-step approaches
5. The redundancy identification using reasoning-specific sequences (Equations 4 and 6) is ad-hoc without justification for why these specific metrics capture reasoning-irrelevant parameters

**Questions:**

1. What is the total wall-clock time and compute cost for the complete PTL process compared to (a) training a smaller model from scratch, and (b) single-step pruning with equivalent recovery training time?
2. How are σ_neuron and σ_layer determined for each model? Is there a systematic approach or does it require extensive grid search? How sensitive is the final performance to these choices?
3. Why does performance sometimes improve in early pruning iterations (e.g., GSM8K: 70.0% → 77.6% for Gemma2)? Is this due to regularization effects or the recovery training data?
4. The redundant parameter identification uses reasoning-specific sequences. How does this affect the model's performance on non-reasoning tasks? Would using mixed-task data for identification preserve broader capabilities?

---

> ### Author Response · Authors · 2025-11-24
> **Response to Reviewer kkhn (Part I)**
>
> Dear Reviewer kkhn,
>
> Thank you for your insightful reviews and comments. We appreciate the time and effort you have put into providing valuable feedback. We would like to address your concerns as follows:
>
>
> > Concern #1: Novelty
>
> We appreciate the opportunity to clarify the novelty and contributions of the proposed PTL method.
>
> 1. Novel Parameter Extraction Criterion
>
> Prior works rely heavily on external data to estimate parameter importance. For example, [1] and [2] use gradient-based criteria, which require labeled data; [3] uses activations and depends on access to training data — both approaches are impractical for LLMs. Additionally, [4] uses parameter magnitude as the pruning criterion, which has been shown to be ineffective in the context of LLMs [5][6].
>
> In contrast, our method introduces a neuron detection mechanism based on influence in embeddings, which captures the semantic contribution of neurons. This approach is intuitive, data-efficient, and does not require ground-truth labels, making it particularly suitable for LLMs where labeled data may be unavailable or expensive to obtain. Our experiments demonstrate that this criterion is both effective and robust.
>
> 2. Significantly More Challenging Task
>
> Most previous pruning studies have focused on CNNs and image classification tasks [1][2][3][4], which are considerably simpler than generation tasks involving LLMs. Furthermore, very few works have explored pruning for generation or reasoning tasks in LLMs [7][8][9].
>
> In contrast, we apply PTL to complex reasoning tasks in LLMs and successfully compress the model to 60% of its original size. This level of compression, while maintaining strong performance on generation tasks, represents a substantial advancement over prior pruning efforts and showcases the practical strength of our approach.
>
> 3. Structured and Deployment-Friendly Pruning
>
> Unlike prior methods that focus on unstructured sparsity — which often does not translate to real parameter reduction and complicates downstream fine-tuning — PTL performs structured pruning with careful architectural design:
>
> - The same number of neurons are pruned across layers, resulting in a nested structure.
> - The resulting model is compatible with efficient inference frameworks such as vLLM.
>
> In contrast, many previous works either:
>
> - Rely on unstructured sparsity (which is hard to leverage efficiently in practice), or
> - Use structure pruning but produce architectures that are not proportionally reduced [10], limiting their practical utility.
>
> We hope this clarification highlights the practicality, novelty, and technical significance of PTL in the context of pruning large-scale generative models.
>
> ---
>
> > Concern #2: Adopt similar multi-step pruning method on baseline
>
> We conducted experiments on ShortGPT using the same setup, where the model is trained after each pruning step. In the case of ShortGPT, we apply layer-wise pruning.
>
> | PTL on shortgpt             | GSM8K | Minerva Math | MATH-500 |
> | --------------------------- | ----- | ------------ | -------- |
> | Gemma2 9B                   | 70.0  | 29.1         | 26.9     |
> | ShortGPT+Iterative Training | 59.8  | 21.2         | 20.7     |
> | PTL                         | 70.6  | 26.4         | 26.2     |
> | LLama3 8B                   | 54.7  | 16.0         | 14.6     |
> | ShortGPT+Iterative Training | 45.3  | 16.8         | 15.7     |
> | PTL                         | 52.5  | 18.5         | 16.1     |
>
> These results demonstrate that even when applying multi-step (iterative) training, PTL consistently and significantly outperforms the baselines across all tasks and model backbones.

---

> ### Author Response · Authors · 2025-11-24
> **Response to Reviewer kkhn (Part II)**
>
> > Concern #3: Theory analysis
>
> Although both PTL and prune-once rely on gradient descent to recover performance after pruning, their effectiveness differs significantly due to the **magnitude and frequency of model perturbations** introduced before each tuning stage. PTL adopts a **gradual pruning strategy**, introducing only small changes at each step, which makes recovery easier, more stable, and more effective.
>
> Let the model be a function $f_\theta : \mathcal{X} \rightarrow \mathcal{Y}$, parameterized by $\theta$. In the prune-once setting, a large subset of parameters is removed all at once. This can be characterized as:
> $\theta' = \theta - \Delta \theta$
> This large pruning step induces a significant shift in the model’s outputs:
> $\mathbb{E}_{x \sim \mathcal{D}} \left[ | f{\theta}(x) - f{\theta'}(x) | \right]$
>
> Such a shift can be **too large for gradient descent to effectively recover from**, especially when fine-tuning time is limited. The size of $| \Delta \theta |$ is typically large and may include entire neurons, layers, or attention heads. Even when the removed weights have low magnitude, their absence can cause **nonlinear disruptions** to the model’s internal representations. These changes can push the model into a **poorly conditioned region** of the loss landscape, where gradients become less informative or even misleading.
>
> Moreover, the relationship between sparsity and performance is **highly nonlinear**. Small pruning levels (e.g., <5%) tend to have minimal effect, but once a **critical threshold** is passed (often around 20–30%), performance drops sharply. This degradation behaves more like a **cliff** than a slope. One-shot pruning often crosses this threshold in a single step, leading to catastrophic functional damage that gradient descent alone cannot reverse.
>
> In contrast, PTL performs $T$ small prune–tune cycles. At each step $t$, only a small fraction $\epsilon \ll 1$ of weights is removed:
> $\theta_t = \text{prune}(\theta_{t-1}, \epsilon)$
> followed by fine-tuning:
> $\theta_t = \text{tune}(\theta_t, \mathcal{D}_{\text{tune}})$
>
> Because each update $| \theta_t - \theta_{t-1} |$ is small, the corresponding change in model outputs is also small. Assuming $f$ is Lipschitz continuous in $\theta$ with constant $L$, we can bound the functional shift:
> $| f_{\theta_t}(x) - f_{\theta_{t-1}}(x) | \leq L | \theta_t - \theta_{t-1} |$
> This gives a per-step output shift on the order of $\mathcal{O}(\epsilon)$. Summing over all $T$ steps:
> $\sum_{t=1}^T | f_{\theta_t}(x) - f_{\theta_{t-1}}(x) | \ll | f_\theta(x) - f_{\theta'}(x) |$
> Thus, PTL keeps the model within a **recoverable regime** throughout the pruning process.
>
> This gradual approach brings several benefits. Because the model remains close to its previous optimum at each step, fine-tuning is **locally effective**: gradients are informative, learning rates remain stable, and the model avoids catastrophic forgetting. Each prune–tune cycle acts as a **local recovery**, rather than a global repair.
>
> In practice, this means PTL behaves as if it enforces a **trust-region constraint**:
> $| \theta_t - \theta_{t-1} | \leq \delta$
> This constraint stabilizes training by ensuring the model remains in a region of parameter space where it behaves predictably and remains aligned with its pre-pruned behavior.
>
> Empirical results support this theoretical intuition. As shown in Section 4.2, PTL retains over 99% of original task accuracy at 50% sparsity, while prune-once suffers significant degradation. Additionally, stability metrics such as output shift norms are consistently lower for PTL, confirming that its updates cause smaller functional perturbations.
>
> **In conclusion**, while both PTL and prune-once rely on gradient descent, PTL succeeds by **controlling the magnitude and frequency of pruning-induced perturbations**. By pruning incrementally and tuning frequently, it keeps the model in a **recoverable and stable region** of the loss landscape at all times. This leads to smaller distributional shifts, faster and more reliable recovery, and significantly better final performance.

---

> ### Author Response · Authors · 2025-11-24
> **Response to Reviewer kkhn (Part III)**
>
> > Concern #4: Details
>
> 1. Time
>
> As we adopt parallel neuron detection method, the neuron detection and pruning is quite fast, especially compared to the training time, normally less than one minute for one step. Therefore, the recovery time reported in the Table is the total time.
>
> Furthermore, it is impossible for us to train a small model from scratch, which is super costly. However, compared to other pruning methods that aim to preserve the original model’s performance, our approach is **significantly more efficient.** For example, [11] requires nearly 100 billion tokens to compress a model to half its original size, a process that would take several months on 8 A100 GPUs. In contrast, our method, PTL, achieves comparable compression in just 20 hours of training, reducing the model size by half while maintaining strong performance. This highlights the efficiency and practicality of our approach.
>
> 2. How to select the threshold.
>
> In fact, our method produces a ranked order of neuron importance rather than relying on a fixed threshold. We remove the neurons with the lowest importance scores—for examole, the bottom 5%. Therefore, there is no predefined or absolute threshold involved in the pruning process.
>
> 3. GSM8K improvement
>
> Thanks for pointing this out. We think this is because of the recovery training data. To further explore the influence of training data, we conducted a comparison between training the full model and training the pruned model in each iteration of PTL. Our findings show that training on the full dataset indeed improves performance, demonstrating the validity and high quality of the dataset. However, we also observe that the pruned model in each PTL iteration achieves performance that is comparable to the full model, despite having a significantly smaller size. This highlights the effectiveness of our pruning method, which maintains competitive performance while reducing model complexity.
>
> ### Reasoning Benchmark
>
> | Model    | GSM8K | MATH | Size |
>
> |--------------|--------|------|------|
>
> | Baseline   | 70.0  | 29.6 | 9.2B |
>
> | CoT Full 1  | 77.6  | 30.0 | 9.2B |
>
> | CoT Full 2  | 78.3  | 31.3 | 9.2B |
>
> | CoT Full 3  | 77.2  | 31.0 | 9.2B |
>
> | CoT Full 4  | 78.1  | 31.3 | 9.2B |
>
> | CoT Full 5  | 77.1  | 31.6 | 9.2B |
>
> | CoT Full 6  | 75.7  | 32.0 | 9.2B |
>
> | CoT Full 7  | 78.0  | 32.5 | 9.2B |
>
> | CoT Full 8  | 75.1  | 31.1 | 9.2B |
>
> ### Pruning Model — Reasoning Benchmark
>
> | Model       | GSM8K | MATH | Size |
>
> |-------------------|--------|------|------|
>
> | Baseline     | 70.0  | 29.6 | 9.2B |
>
> | PTL Iteration 1  | 75.6  | 29.0 | 8.8B |
>
> | PTL Iteration 2  | 77.6  | 29.4 | 8.4B |
>
> | PTL Iteration 3  | 74.1  | 28.9 | 7.9B |
>
> | PTL Iteration 4  | 74.4  | 28.3 | 7.5B |
>
> | PTL Iteration 5  | 74.7  | 27.9 | 7.0B |
>
> | PTL Iteration 6  | 73.4  | 27.4 | 6.6B |
>
> | PTL Iteration 7  | 69.3  | 26.3 | 6.2B |
>
> | PTL Iteration 8  | 73.3  | 27.5 | 5.8B |
>
> 4. Non-reasoning tasks
>
> We evaluate the pruned model on SQuAD, a commonsense reasoning benchmark. The results are shown below. Interestingly, although the models were only recovered using the math reasoning task, their performance on other reasoning tasks—including SQuAD—remains well-preserved, demonstrating the generalization ability of the compacted models.
>
> |             | Gemma2-8B | PTL-Gemma2-5 | Llama3-8B | PTL-Llama3-5B |
> | ----------- | --------- | ------------ | --------- | ------------- |
> | Exact Match | 27.4      | 28.2         | 27.3      | 37.2          |
> | F1          | 34.8      | 32.9         | 33.6      | 40.6          |
>
> [1] Learning effective pruning at initialization from iterative pruning. arXiv 2024
>
> [2] Pruning neural networks without any data by iteratively conserving synaptic flow. NeurIPS 2020
>
> [3] DropNet: Reducing Neural Network Complexity via Iterative Pruning. ICML 2020
>
> [4] Optimizing Learning Rate Schedules for Iterative Pruning of Deep Neural Networks. TMLR 2023
>
> [5] Language-Specific Neurons: The Key to Multilingual Capabilities in Large Language Models. ACL 2024
>
> [6] The Rise of Parameter Specialization for Knowledge Storage in Large Language Models. arXiv 2025
>
> [7] SliceGPT: Compress Large Language Models by Deleting Rows and Columns. ICLR 2024
>
> [8] LLM-Pruner: On the Structural Pruning of Large Language Models. NeurIPS 2023
>
> [9] ShortGPT: Layers in Large Language Models are More Redundant Than You Expect. arXiv 2024
>
> [10] What is the State of Neural Network Pruning? MLSys 2020
>
> [11] Compact Language Models via Pruning and Knowledge Distillation. NeurIPS 2024

---

### Official Review · Reviewer_CbPM · 2025-10-29

**Soundness:** 3
**Presentation:** 3
**Contribution:** 2
**Rating:** 4
**Confidence:** 3

**Summary:**

This paper proposes a progressive compression method for Large Language Models (LLMs) called the Prune-Tune Loop (PTL). This method gradually removes redundant parameters (neurons or layers) through multiple iterative "pruning-fine-tuning" cycles and leverages Chain-of-Thought (CoT) data or reinforcement learning for performance recovery. Experiments were conducted on models such as Llama3-8B, Gemma2-9B, and Qwen2.5-7B. The experimental results show that with only lightweight subsequent training, PTL can compress an LLM to nearly half of its original size while maintaining performance comparable to the original model on reasoning tasks.

**Strengths:**

1. The paper is well-written and easy to follow.
2. The core idea of the PTL method—progressive compression to avoid sudden performance degradation—is reasonable. Compared with one-time pruning (e.g., Prune-Once), this method demonstrates better performance recovery.
3. The paper conducts extensive tests on multiple open-source models and benchmarks, including models such as Llama3-8B, Qwen2.5-7B, and Gemma2-9B, as well as mathematical reasoning benchmarks (GSM8K, Minerva Math, MATH-500) and the code generation benchmark (MBPP). The results show that the compressed model achieves performance close to that of the original model, while significantly optimizing FLOPs and inference speed.

**Weaknesses:**

1. In my view, the biggest issue with this paper is lack of novelty. Progressive pruning and fine-tuning are not novel concepts—they were widely applied during the era of Convolutional Neural Networks. For instance, the idea of progressive pruning and fine-tuning was proposed quite early in [1]. The authors merely extended progressive pruning and fine-tuning to the pruning of LLMs, and the continuous pre-training and reinforcement learning methods they adopted are also existing algorithms.
2. The reinforcement learning training experiments were conducted on eight 140GB NVIDIA H200 GPUs, and the fine-tuning of LLMs at the 7B scale takes 64 hours, which still constitutes a non-negligible cost. If scaled to LLMs of larger scales (e.g., the 70B scale), the fine-tuning cost will be even higher.
3. The hypothesis that only pruning the FFN does not affect attention is based on the existing conclusion that "reasoning mainly depends on self-attention," and there is a lack of an experiment to verify the above hypothesis.
4. Typos: Line 174, Reinformence Learning->Reinforcement Learning

[1] Molchanov P, Tyree S, Karras T, et al. Pruning Convolutional Neural Networks for Resource Efficient Inference [C]. International Conference on Learning Representations. 2017.

**Questions:**

1. How are the thresholds for redundant parameter extraction ($σ_{neuron}$ and $σ_{layer}$) selected? The authors seem to have omitted this detail.
2. In the reinforcement learning-based recovery process, why did other baselines fail (achieving 0% accuracy)? Is it due to model collapse caused by structural changes, or issues with training configurations?

---

> ### Author Response · Authors · 2025-11-24
> **Response to Reviewer CbPM**
>
> We would like to address your concerns as follows
>
> > Concern #1: Novelty
>
> 1. Novel Parameter Extraction Criterion
>
> Prior works rely heavily on external data to estimate parameter importance. For example, [1] and [2] use gradient-based criteria, which require labeled data; [3] uses activations and depends on access to training data — both approaches are impractical for LLMs. Additionally, [4] uses parameter magnitude as the pruning criterion, which has been shown to be ineffective in the context of LLMs [5][6].
>
> In contrast, our method introduces a neuron detection mechanism based on influence in embeddings, which captures the semantic contribution of neurons. This approach is intuitive, data-efficient, and does not require ground-truth labels, making it particularly suitable for LLMs where labeled data may be unavailable or expensive to obtain. Our experiments demonstrate that this criterion is both effective and robust.
>
> 2. Significantly More Challenging Task
>
> Most previous pruning studies have focused on CNNs and image classification tasks [1][2][3][4], which are considerably simpler than generation tasks involving LLMs. Furthermore, very few works have explored pruning for generation or reasoning tasks in LLMs [7][8][9].
>
> In contrast, we apply PTL to complex reasoning tasks in LLMs and successfully compress the model to 60% of its original size. This level of compression, while maintaining strong performance on generation tasks, represents a substantial advancement over prior pruning efforts and showcases the practical strength of our approach.
>
> 3. Structured and Deployment-Friendly Pruning
>
> Unlike prior methods that focus on unstructured sparsity — which often does not translate to real parameter reduction and complicates downstream fine-tuning — PTL performs structured pruning with careful architectural design:
>
> - The same number of neurons are pruned across layers, resulting in a nested structure.
> - The resulting model is compatible with efficient inference frameworks such as vLLM.
>
> In contrast, many previous works either:
>
> - Rely on unstructured sparsity (which is hard to leverage efficiently in practice), or
> - Use structure pruning but produce architectures that are not proportionally reduced [10], limiting their practical utility.
>
> We hope this clarification highlights the practicality, novelty, and technical significance of PTL in the context of pruning large-scale generative models.
>
> ---
> > Concern #2: Training cost
>
> Yes, the training cost may appear high when considering only the training time. However, compared to other pruning methods that aim to preserve the original model’s performance, our approach is **significantly more efficient.** For example, [11] requires nearly 100 billion tokens to compress a model to half its original size, a process that would take several months on 8 A100 GPUs. In contrast, our method, PTL, achieves comparable compression in just 20 hours of training, reducing the model size by half while maintaining strong performance. This highlights the efficiency and practicality of our approach.
>
> ---
>
> > Concern #3: Missing details
>
> 1. How to select the threshold.
>
> In fact, our method produces a ranked order of neuron importance rather than relying on a fixed threshold. We remove the neurons with the lowest importance scores—for examole, the bottom 5%. Therefore, there is no predefined or absolute threshold involved in the pruning process.
>
> 2. 0% for baselines in the RL setting.
>
> This happens because the model completely fails, meaning it cannot generate any meaningful responses. As a result, the RL training process also fails.
>
> 3. Prune neurons in the attention layer.
>
> We explored this approach, but found that pruning even just one or two neurons within the attention structure causes the model to completely fail. Moreover, due to the multi-head nature of the attention mechanism, we must either prune the same number of neurons in each head or prune entire heads. In both cases, the model fails to function properly.
>
> ---
>
> [1] Learning effective pruning at initialization from iterative pruning. arXiv 2024
>
> [2] Pruning neural networks without any data by iteratively conserving synaptic flow. NeurIPS 2020
>
> [3] DropNet: Reducing Neural Network Complexity via Iterative Pruning. ICML 2020
>
> [4] Optimizing Learning Rate Schedules for Iterative Pruning of Deep Neural Networks. TMLR 2023
>
> [5] Language-Specific Neurons: The Key to Multilingual Capabilities in Large Language Models. ACL 2024
>
> [6] The Rise of Parameter Specialization for Knowledge Storage in Large Language Models. arXiv 2025
>
> [7] SliceGPT: Compress Large Language Models by Deleting Rows and Columns. ICLR 2024
>
> [8] LLM-Pruner: On the Structural Pruning of Large Language Models. NeurIPS 2023
>
> [9] ShortGPT: Layers in Large Language Models are More Redundant Than You Expect. arXiv 2024
>
> [10] What is the State of Neural Network Pruning? MLSys 2020
>
> [11] Compact Language Models via Pruning and Knowledge Distillation. NeurIPS 2024

---

### Official Review · Reviewer_kLiY · 2025-11-01

**Soundness:** 3
**Presentation:** 3
**Contribution:** 2
**Rating:** 4
**Confidence:** 4

**Summary:**

This paper identifies that standard one-shot pruning methods cause catastrophic failure in mathematical and code reasoning tasks. To solve this, the authors propose an iterative compression framework Prune-Tune Loop (PTL) . In each fine-grained iteration, the model is pruned by removing redundant reasoning parameters and then recovered via lightweight post-training (using Continual Pre-training on Chain-of-Thought data or Reinforcement Learning). Experiments on models like Llama3-8B and Gemma2-9B show that PTL can compress models to ~60% of their original size while maintaining reasoning accuracy.

**Strengths:**

- The pruning and tuning loop makes sense to me. It is well-motivated to recover the model's ability after pruning.

- The ablation studies on pruning step size, iterations, and order provide valuable insights into the method's behavior and stability.

- The comparison against the one-shot pruning methods demonstrates their effectiveness in ability recovery.

**Weaknesses:**

- I think a significant weakness is the lack of discussion on scalability. The method is only validated on models up to 9B parameters. Given its reliance on multiple rounds of post-training, the computational cost of larger models (e.g., 70B+) remains a critical question.

- The method appears sensitive and requires per-model tuning. The need for different recovery strategies (RL for Qwen vs. Continual Pre-training for others) and the significant performance variance with different pruning step sizes (5% vs. 20%) suggest PTL is not a plug-and-play solution.

**Questions:**

- What is your estimate of the computational cost of applying PTL to a much larger model like a 70B model?

- While the focus of this work is math reasoning, do you believe this framework, without dedicated modification, can be adapted to other domains like logical deduction?

---

> ### Author Response · Authors · 2025-11-24
> **Response to Reviewer kLiY**
>
> Dear Reviewer kLiY,
>
> Thank you for your insightful reviews and comments. We appreciate the time and effort you have put into providing valuable feedback. We would like to address your concerns as follows:
>
> > Concern #1: Scalability
>
> Yes, we fully agree on the importance of evaluating our method on larger models, such as those with 70B parameters. However, due to limited computational resources, we are currently unable to conduct experiments at that scale. For instance, training a 70B model typically requires at least 32 A100 GPUs, and even then, it operates nearly ten times slower than training an 8B model with 8 A100 GPUs. Given these constraints, experiments on a 70B model are not feasible for us at this time.
>
> That said, we appreciate the reviewer’s suggestion to investigate the scalability of our method. In response, we conducted additional experiments using a larger model, Gemma3-12B, to assess the performance of our method at a greater scale. The results are as follows:
>
> | PTL on Gemma3-12B | GSM8K | Minerva Math | MATH-500 |
> | ----------------- | ----- | ------------ | -------- |
> | Original          | 89.2  | 34.9         | 37.0     |
> | PTL-9B            | 86.6  | 32.1         | 34.5     |
>
> These results indicate that our method, PTL, remains effective and scalable when applied to a 12B-parameter model. We believe this provides meaningful evidence of the method’s potential to generalize to even larger models, pending sufficient resources in the future.
>
> ---
>
> > Concern #2: PTL is plug-and-play
>
> We explore different settings, such as varying the pruning steps, to conduct ablation analysis and demonstrate the generalizability of our method.
>
> For the recovery method, we also test on Qwen for the continual pre-training recovery method. Experimental results show that the Qwen series models are particularly difficult to recover through continual pre-training. We hypothesize that this is because Qwen models have already been extensively trained on datasets that are either overlapping with or highly similar to the evaluation benchmarks. As a result, their performance is already near-optimal, and further recovery using only standard open-source datasets is insufficient to regain their original capabilities.
>
> | Pruning Model   | ReasoningBenchmark |      |      |
> | --------------- | ------------------ | ---- | ---- |
> |                 | GSM8K              | MATH | Size |
> | Baseline        | 83.7               | 38.7 | 7.6B |
> | PTL Iteration 1 | 75.7               | 38.2 | 7.2B |
> | PTL Iteration 2 | 75.7               | 34.2 | 6.8B |
> | PTL Iteration 3 | 75.1               | 32.9 | 6.4B |
> | PTL Iteration 4 | 69.7               | 31.7 | 6.0B |
>
> Instead, as Qwen is not trained by RL, it can be used to recover its performance. In fact, retraining pruned models on new data is a well-established and widely adopted approach for performance recovery [1][2][3]. That said, we acknowledge that recovery via RL—particularly for tasks with unverifiable or open-ended outputs—warrants further investigation.
>
> In summary, no single method can be expected to perform optimally across all possible settings and scenarios. However, our comprehensive ablation analysis—including variations in pruning steps, training corpus, pruning order, and task types—demonstrates the strong generalizability and robustness of our approach.
>
> ---
>
> [1] LLM-Pruner: On the Structural Pruning of Large Language Models. NeurIPS 2023
>
>
>
> [2] Compact Language Models via Pruning and Knowledge Distillation. Arxiv 2024
>
>
>
> [3] SliceGPT: Compress Large Language Models by Deleting Rows and Columns. ICLR 2024

---

### Meta-Review · Area_Chair_2CFb · 2026-01-07

**Summary:**

This work considers progressive model compression in the context of reasoning problems, iteratively pruning redundant parameters during post-training. Compared to baseline models, the approach is shown to perform better and results are validated across a range of hyperparameter settings.

The concerns raised by the reviewers are relatively broad, the main ones being:
1/ The method is only validated on relatively small models (up to 9B parameters)
2/ The proposed approach relies on progressive pruning and fine-tuning, which are not novel concept
3/ Lack of theoretical analysis and no new insights why gradual removal preserves reasoning capabilities.
4/ Missing baselines such as  iterative magnitude pruning, gradual sparsity methods, or recent structured pruning techniques that use similar multi-step approaches.

**Reviewer Concerns:**

The authors provided very detailed  and informative responses, which addressed a large fraction of the concerns raised:

1/ Authors provided additional supporting evidence by conducting new experiments. The details provided are convincing and suggests the method had the potential to generalise to larger models.
2/ The authors provided an extensive and detailed response outlining the novel aspects of their work, convincing me that there is enough material.
3/ The authors provided a detailed theoretical analysis during the rebuttal.

However, they did not attempt to elaborate on lacking insights, nor did they replied to the concern 4/ above raised by reviewer kkhn. It actually appears that they missed a few of the concerns raised (weaknesses 3-5 and questions 1-2 if I am not mistaken) and only addressed a subset. This is unfortunate.

**Reviewer Scores:**

Most reviewers voted initially for rejection, but the authors provided convincing arguments and additional details to resolve most issues. Unfortunately, it appears the missed a number of important concerns raised by one of the reviewer, for which they did not provide a rebuttal. I would have expected a fraction of the scores to increase post rebuttal making this paper close to borderline accept, but given that a number of concerns remained unanswered I cannot recommend acceptance.

---

### Decision · Program_Chairs · 2026-01-26

Reject